# Varieties of Explainable Agency

## Pat Langley

Institute for the Study of Learning and Expertise,
2164 Staunton Court, Palo Alto, CA 94306 USA

Department of Computer Science, University of Auckland,
Private Bag 92019, Auckland 1142 NZ

## Abstract

In this paper, I discuss some varieties of explanation that can arise in intelligent agents. I distinguish between process accounts, which address the detailed decisions made during heuristic search, and preference accounts, which clarify the ordering of alternatives independent of how they were generated. I also hypothesize which types of users will appreciate which types of explanation. In addition, I discuss three facets of multi-step decision making – conceptual inference, plan generation, and plan execution – in which explanations can arise. I also consider alternative ways to present questions to agents and for them provide their answers.

## 1 Introduction

Intelligent systems are becoming more widely adopted for critical tasks like driving cars and controlling military robots. Our increased reliance on such devices has led to concerns about the interpretability of their complex behavior. Before we can fully trust such autonomous agents, they must be able to explain their decisions so that we can gain insight into their operation. There is now a substantial literature on explanation in systems that learn from experience, but it has focused on tasks like object recognition and reactive control, typically using opaque encodings of expertise.

However, we also need research on explanation for more complex tasks that involve multi-step decision making, such as the generation and execution of plans. Approaches to these problems rely on high-level representations that are themselves easily interpreted, but challenges arise in communicating solutions that combine these elements and the reasons they were chosen. In this paper, I focus on such settings. Some work on explanation, especially with opaque models, has dealt with post hoc rationalizations of behavior, rather than the actual reasons for it. In the pages that follow, I limit my discussion to the latter. Moreover, I will focus on *self explanations*, that is, the reasons the explaining agent carried out a certain activity. Elsewhere (Langley, 2019), I have referred to this ability as *explainable agency*. This problem is arguably less challenging than postulating the reasons that another agent behaved as it did, sometimes called *plan recognition*, as the system can store and access traces of its own decision making.

We can specify the task of explainable agency in generic terms. Given domain knowledge for generating task solutions and criteria for evaluating candidates, the agent carries out search to find one or more solutions. After generating, and possibly executing, these solutions, a human asks the agent to justify its decisions, at which point it must clarify its reasoning in comprehensible terms. One example involves an intelligent robot that plans and executes a reconnaissance mission, after which it takes part in an 'after-action review' where it answers questions from a human supervisor. There has been some research on such *explainable planning* (Fox et al., 2017; Smith, 2012; Zhang et al., 2017), but we need more effort devoted this important topic.

In the next section, I distinguish between two forms of self explanation, identify component abilities they require, and citing relevant research. I also propose two hypotheses about when each type of account will be most useful. After this, I discuss three types of content over which one can generate explanations, along with alternative ways to pose questions and present answers. In closing, I review the essay's main points and reiterate the need for substantially additional research on the topic of explainable agency.

## 2 Forms of Self Explanation

Before the research community can develop computational methods for self explanation, it must first establish which aspects of decision making to elucidate. I maintain that there are two primary forms of explanation, which I attempt to characterize in this section. In each case, I offer some intuitions, define the task in terms of inputs and outputs, and discuss components that appear necessary to carry it out.

### 2.1 Process Accounts

The first form of self explanation focuses on the *processes* that led a system to generate its plans or other mental structures. This view revolves around the widespread assumption, which had its origins in the earliest days of artificial intelligence, that complex cognition requires heuristic search through a problem space (Newell and Simon, 1976). This assumes that the recipients of explanations are interested in details about how the system carried out that search, including which alternatives it considered, why it decided to pursue some in favor of others, and even when it decided to change its mind (e.g., by deciding to backtrack).

We can specify the generic task of explaining the processing that produced solutions as:

- *Given:* Knowledge defining a space of possible solutions;
- *Given:* Criteria for evaluating candidate solutions;
- *Given:* An annotated search tree that includes solutions found for some reasoning task;
- *Given:* A query about why a solution ranks above others;
- *Produce:* An explanation why the solution is preferable.

This task formulation is similar in spirit to the generation of think-aloud protocols (Newell and Simon, 1972), which gave early insights about human problem solving and which led directly to the creation of early AI systems. In this setting, a researcher presents a subject with some problem (e.g., a theorem to prove or a puzzle to solve), asking the subject to talk aloud as he works on it. The researcher records this verbal report, transcribes it, and analyzes it to understand the subject's thinking processes. One important difference is that our explanation task occurs after problem solving is complete. Retrospective reports from in humans far less reliable than on-line protocols, but AI systems have better memories than people, so I will ignore this issue.

Elsewhere (Langley, 2019), I have analyzed the component abilities that appear necessary to support this variety of self explanation. These include ensuring that the intelligent agent can:

- *Generate decision-making content*. When carrying our heuristic search, an agent must consider different nodes and operators, evaluate them, and select one to pursue.
- *Store generated content*. When it makes such decisions, the agent must store and index details about the choices it considered, and why it selected one of them, in an episodic memory or similar repository.
- *Retrieve stored content*. After it has solved a problem, the agent must transform questions into cues that let it retrieve traces of relevant decisions from episodic memory.
- *Communicate retrieved content*. Once it has retrieved this information, it must translate this content into an understandable form and communicate it.

Taken together, these abilities should let an intelligent system not only find solutions to complex problems, but also recount how it managed to uncover them.

The AI literature includes some relevant research on these topics. For instance, work on analogical planning (e.g., Jones and Langley, 2005; Veloso et al., 1995) has addressed storage, indexing, and retrieval, but not for use in self explanation. Some expert systems recorded their reasoning and played them back on request (Clancey, 1983; Swartout et al., 1991), while Johnson (1994) and van Lent et al. (2004) developed agents that carried out military mssions, recorded their decisions, and answered questions about their reasoning. Other related work includes an interactive robot that can give five types of reasons why it cannot carry out a task (Briggs and Scheutz, 2015) and computational models of argument (e.g., Bench-Capon and Dunne, 2007) that explain how alternative conclusions are eiter supported or contradicted by available evidence.

## 2.2 Preference Accounts

The second form of self explanation focuses on the final solutions produced by heuristic search, without concern for how they were found. This view recognizes that there are many different techniques for problem solving. A classic example is that some planning methods chain backward from goal descriptions, whereas others chain forward from the initial state. Similarly, some approaches to constraint satisfaction carry out search through a space of partial variable assignments, whereas others only consider alternatives that have complete assignments. Even within the same framework, different heuristics can guide search independently of the target goals or objective function. Nevertheless, these different systems can arrive at the same solutions by distinct paths, which can be sources of explanation themselves.

As before, we can state this task more precisely in terms of inputs and outputs:

- *Given:* Knowledge defining a space of possible solutions;
- *Given:* Criteria for evaluating candidate solutions;
- *Given:* A ranked set of solutions to some reasoning task;
- *Given:* A query about why a solution ranks above others;
- *Produce:* An explanation why the solution is preferable.

This task is very different from generating think-aloud protocols about the choices considered and selected during heuristic search. Rather, it comes much closer to the task addressed by recommender systems, which often produce a ranked list of candidates for users to consider. Most of these focus on ranking a fixed set of items, such as books, but one can also rank solutions to planning, scheduling, and other tasks that involve multi-step reasoning.

The distinction between process and preference explanations is not a matter of granularity, but whether one cares about *means* of reaching results or about their *quality*. To clarify this point, consider a simplified case-based reasoning system that iteratively retrieves a complete plan from memory, replacing one of the *n* best candidates with a new one if the latter scores better. A process account would store the sequence of candidates considered and explain its final choices in terms of steps in this procedure. In contrast, a preference account would retain only the final set of candidates and explain their ordering in terms of how each one fares on its criteria. An explainable agent should also, when a given candidate is not in the solution set, state why it was (presumably) ranked lower than those included.

Neither does the emphasis on preferences imply that explanation must only deal with complete solution structures. For example, if a planner uses a hierarchical task network to guide its search, then a user should be able to question why it selected one subplan for a given subtask rather than another decomposition. The same idea applies to a system that finds proof trees using monotonic inference rules, where a user may ask why it favored one subproof over another candidate that leads to the same intermediate conclusion. The ability to focus attention on elements of hierarchical solutions does not necessarily mean that explanations must touch on how the solutions were found.

Let us consider the component abilities that seem needed for an intelligent agent to provide such preference explanations. These include the capacity to:

- *Generate and rank solutions*. However the agent solves a problem, it must use explicit, interpretable criteria to place an ordering on them.
- *Compare two ranked solutions*. When asked why one solution was placed before another, the agent must compare their component scores and their combinations.
- *Communicate solution differences*. Once it has noted how the candidates differ, it must convey this information and how it led to their relative rankings.

The details of these abilities will depend on how the scoring and ranking process operates. One common scoring method uses a linear utility function that computes each candidate's score on $k$ features, multiplies each score by a weight, and calculates a weighted sum, then orders candidates by this total. Another scheme uses a lexicographic function, which orders attributes by importance. Candidates are first partitioned based on scores for the first attribute, then ranked within these sets based on the second attribute, and so forth, much as words in a dictionary. The structure of explanations will depend on the technique used to order solutions.

I mentioned earlier the analogy to recommender systems, which often rely on a learned user profile to rank candidate items like books or movies. However, one can use such profiles as heuristics to guide search on complex reasoning tasks and to rank the solutions found in this manner. Rogers et al. (1998) applied this idea to route planning, drawing on a user profile, stated as weights on route features, to find personalized routes in a digital road map. Gervasio et al. (1999) adopted a similar approach to personalized scheduling, invoking a user profile, here weights on schedule features, to evaluate candidates and rank solutions. These two efforts are interesting because the first used best-first search through a space of partial routes, whereas the second used repair-space search through a space of complete schedules. Together, they offer evidence that one can have the same type of preference explanations for radically different search methods.

## 2.3 Two Hypotheses about Explanations

Now that we have identified and characterized two forms of self explanation, we can ask which is them is more useful to humans who interact with intelligent agents. One might argue that process explanations are the natural choice, as providing more details will give greater insight into a system's operation. But one might instead hold that preference accounts are superior, because humans have no need to know how the system found its solutions but only why it ranked the alternatives as it did.

In this paper, I will not take either position, but instead claim that the most appropriate form of explanation depends on the user's aims. This argument assumes that there are two quite different types of users, which leads to two hypotheses. We can state the first as:

- *Hypothesis 1: Process explanations will be favored by researchers interested in the details of heuristic search.*

This conjecture posits that some users care primarily about the process of finding solutions. This group includes cognitive psychologists who want to understand the ways in which an intelligent system mimics, or fails to mimic, a human problem solver. Yet it also includes many AI researchers who are concerned with the detailed operation of their systems, both for debugging purposes and for improving their search mechanisms.

However, not all users of intelligent systems will care about the technical details of their search behavior. This suggests a second conjecture, which we can state as:

- *Hypothesis 2: Preference explanations will be favored by system users interested in the results of heuristic search.*

This group includes end users of autonomous agents who had no role in their development. These are analogous to people who use recommender systems but have little idea how they operate, but who still want to know why they ranked one item as better than another. But it will also include AI researchers, and even psychologists, who are concerned more with criteria used to evaluate solutions than with the search mechanisms that produce them.

## 3 Types of Explanatory Content

We should also consider the types of tasks over which explanations of complex multi-step reasoning can occur. Planning is the most obvious class of domains and the one that has received the most attention in the literature (Fox et al., 2017; Smith, 2012). Clearly, a planning system can support both forms of explanation discussed above. At each stage in the search process, it can store the choices considered, their associated scores, and the alternative selected, along with decisions about when to backtrack. This information will let it answer detailed queries about the search history. Of course, planning systems can also find multiple solutions, rank them, and use their scoring procedure to provide preference explanations instead.

Plan execution is another important arena that supports explainable agency (Johnson, 1994; van Lent et al., 2004). This setting definitely supports process accounts, as the agent must monitor the environment to determine whether the plan is proceeding as expected. Detection of anomalies can be recorded, along with decisions about whether to continue or to revise the plan. The role of preference accounts is less obvious when the plans being executed are fully grounded, as no alternatives are available. However, frameworks that include reactive control constrained by hierarchical task networks (e.g., Choi and Langley, 2018) do allow multiple choices that the agent can rank by value. These will still be local to the particular situation in which the agent is taking action, but they should support preference accounts.

A third area involves conceptual inference, which may not count as agency itself but which certainly supports it. Here the intelligent system uses knowledge to draw conclusions about its situation from information available to it, using either deductive or abductive reasoning. The latter mechanisms clearly support process accounts, as demonstrated by the early work on explainable diagnostic systems (Clancey, 1983; Swartout et al., 1991), which stored traces

of the reasoning chains that led to their conclusions. However, conceptual inference also supports preference explanations, since the system may have criteria for evaluating alternative derivations, such as the length of its reasoning chain or the number of default assumptions. Such accounts are most interesting in settings that involve incomplete information, where the system may find multiple contradictory interpretations of its situation.

## 4  Interaction Modalities

As I have defined them, explainable agents must be able to accept questions about their decision making and answer them in terms a human can understand, but this does not specify the modality used for either input or output. For questions, one obvious alternative is natural language, but this could be very constrained. In the planning context, process-oriented queries might use stock phrases to ask what actions the agent considered, how it scored each one, the expected results, and which one it selected. Each would need to include context about the situation, such as *upon coming to the end of the hallway*, as this will be needed to identify and retrieve relevant decisions. If the agent has generated multiple plans, each question would also specify which one to examine. Questions in natural language about preference accounts could be much simpler, as they need only state which candidates the agent should conttrast, although hierarchical solutions will require some way to specify a subsolution.

Another option would be to ask questions through a graphical interface that displays, for process explanations, the search tree that produced solutions. By clicking on a particular node in this tree, the user would specify both the plan and the situation being addressed at that point (e.g., when the agent has come to a fork in the hallway). A drop-down menu would let the user indicate whether he wants to know about the choices considered at that point, their evaluation scores, or the one selected. Again, for preference explanations, a graphical interface would be much simpler, displaying alternative solutions, their scores on each criterion, and the results of combining them. The user would click on two solutions to compare them or propose his own candidates if he wants to know why the system did not include them. For hierarchical solutions, the interface would hide details initially but let the user drill down when desired to inspect the rankings for subtasks and the reasoning behind them.

After it has accessed the relevant information, the agent must respond to the question in terms the user will understand. For process explanations, natural language answers can rely on templates that are instantiated with relevant domain terms. Thus, given a query about what choices it entertained upon reaching the branch in the hallway, it might say *I considered turning left and turning right*, which describe the two actions available in that situation. For preference accounts, the agent could simply show the values and weights of solution criteria, along with the calculation that combined them, for the contrasted candidates. Here different templates would be needed for alternative types of ranking methods. Graphical interfaces offer another way to answer process-related questions, say by highlighting selected choices in the search tree, or preference-oriented ones, say by graphing the scores and weights of solution criteria. Systems that combine natural language and graphical interactions may be desirable, as some users could have an easier time understanding textual explanations, while others could instead favor diagrams and graphs.

## 5  Closing Remarks

In this paper, I reviewed the notion of explainable agents, which answer questions about the reasoning behind their complex decision making. I distinguished between two varieties of explanation, one that focuses on the process of finding solutions and another that addresses only the ranking of these candidates. I also proposed two hypotheses: that researchers interested in details of heuristic search will favor process accounts, while end users will be more interested in preference accounts. After this, I argued that plan generation, plan execution, and conceptual inference all support both types of explanation, but that execution poses some difficulties for preference accounts. Finally, I discussed different modalities that humans might use to present explanation-related questions, as well as ones that our agents might use to answer them.

Research on explainable agency should have high priority within both planning circies and the broader AI community. Intelligent agents must be able to explain their reasoning in terms that are comprehensible by humans and that are relevant to their aims. System users will have different priorities, focus on distinct problem types, and favor different modalities, and we need frameworks that support their full range of preferences. However, the first steps should be to design, implement, and demonstrate examples of explainable agents that exhibit each of the abilities identified in the analysis presented here. The experience gained through these efforts will reveal additional challenges that the research community must overcome to develop truly understandable and trustworthy intelligent systems.

## Acknowledgements

This analysis presented in this paper was supported by a contract from the Army Research Laboratory and by Grant N00014-17-1-2434 from the Office of Naval Research, neither of which are responsible for its contents.

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
