# OpenReview forum: "Varieties of Explainable Agency"
_icaps-conference.org/ICAPS/2019/Workshop/XAIP — XAIP 2019_

### Official Review · AnonReviewer1 · 2019-05-09
**A position/survey paper that is behind the state of the art**

**Rating:** 2
**Confidence:** 3

**Review:**

The paper discusses different aspects of explanations, particularly in the context of sequential decision making. Many of these issues have been discussed before (e.g. some in the author's own previous work [IAAI 2017]) and indeed algorithms exist that have begun addressing aspects of these problems since then. Unfortunately, the most recent works referred to in the paper (other than the author's own) are mostly from ages ago. In that context, I am not sure that the paper advances the state of our understanding of the different aspects of XAIP. The contribution would have been much more appealing if it was able to contextualize the same ideas in light of the state of the art.

The difference between process accounts and preference accounts is quite stark in recent literature -- this is better described as model-based algorithm-agnostic explanations or not. Most of the explanations/visualization work in the planning literature has been about process, while all the recent work [Fox et al. 2017] and its follow-ups, and model reconciliation [Chakraborti et al. 2017] and its follow-ups have been model-based algorithm-agnostic. Personally, I think "preference accounts" is a misnomer. Section 2.1 refers to some existing examples of process accounts. I would have loved to see the same for ideas expressed in Section 2.2 on how they manifest themselves in different forms in recent efforts in the XAIP community.

> I couldn't understand the point of "conceptual inference" as a separate concept in Section 3. Surely this happens as part of both "plan generation" and "plan execution"?

> Similar to the concern above, I would refer to the visualization and explanation attempts in the most recent UISP workshops at ICAPS for some examples of the discussion in Section 4.

> (For the sake of completeness) The author makes a note that explanations being retrospective doesn't make much of a difference to the explanation process for "process accounts". In the context of "preference accounts" in the cases of plan generation versus execution, retrospection actually plays a role. Such as not needing to explain that something is executable if the plan has already been executed.

---

### Official Review · AnonReviewer4 · 2019-05-13
**possibly interesting position paper for discussion; SOA coverage weak**

**Rating:** 3
**Confidence:** 2

**Review:**

The paper outlines XAIP "from the ground up", nicely structuring the field -- what it is or should be, what the issues are, what possible solutions could look like -- from a particular point of view.

The discussion remains at a fairly abstract/shallow level. But that is natural for a paper of this kind.

What bothers me more is the lack of discussion of recent works in XAIP, reg model reconciliation etc. If the paper's ambition is to structure XAIP in its entirety along useful distinction lines  (and this is what the text reads like), then where are all the other works that somehow relate to this area, and that fly that flag? It's far from clear how the described delineation would encompass these, or classify them as "something else".

I still think this could make an interesting presentation for discussion at the workshop; but only if the author makes an effort to align his concepts with (or compare the to, or differentiate them from) the various other viewpoints on XAIP in the literature. To name just two specific questions: How does this relate to the range of isues outlined by Fox et al 2017? How does it relate to the prominent work line on model reconciliation? I urge the author to address this and related questions if accepted.

---

### Official Review · AnonReviewer2 · 2019-05-13
**Explanation categorization needs more context, detail**

**Rating:** 4
**Confidence:** 2

**Review:**

The primary contribution of this work is a discussion of two different types of self explanation, referred to as "process" and "preference" accounts. These go toward breaking down the different ways humans use and view explanations, which is important work that remains incomplete. Early work such as Leake's Evaluating Explanations and recent work such as Hoffman's Explaining Explanations could possibly provide useful context for the distinction advanced, which appears to be missing. The work appears to be fairly early, and leaves certain open questions:

- What dimension do these types fall along? Purpose? Broad content category?
- Is this dichotomy exhaustive?
- How does this dimension relate to other previously identified dimensions?
- Is there any intersection?

Answers to such questions would help the conceptual categories advanced by the author to take hold as distinctions useful to other researchers.

The motivation of this work remains unclear. You state that further research into "explainable agency" is needed, and motivate this. However, the statement that you will distinguish two forms of explanation is then unmotivated as the specific research performed.

Comparison between "process" and "preference" accounts appears in the "preference" account subsection. Some of this should likely be hinted at earlier to give motivation, and the rest deserves its own subsection.

My primary criticism of this work stems from the fact that I want to see a paper entitled "Varieties of Explainable Agency" provide a stronger taxonomy that can help clear up the current confusion about what explanation means, and the large variability in the work that it covers. Discussing what users want from explanations is a good first step, but many different metrics are currently applied in the explanation space, and there's little understanding of which are most valuable for what purposes. I'd like to see this work extended to cover additional dimensions, metrics, and more. That being said, this is still a useful contribution to the literature, and an important topic for discussion. I would implore the authors to provide more clarity to the subject.

---

### Decision · Program_Chairs · 2019-05-15

**Decision:**

Accept

**Comment:**

While the reviewers do not fully agree on the decision, in the spirit of making the workshop a venue for discussion and feedback we decided to reject only those papers with strong reject votes.

Please address all review criticism as best possible for the final paper version and its presentation at the workshop. Looking forward to discuss your work at the workshop!

---

> ### Author Response · Authors · 2019-05-16
> **Request for relevant references**
>
> I'm pleased that you have decided to accept my submission to the workshop.
> I will do my best to incorporate the reviewers' suggestions, but it would help
> a lot if those who felt the paper did not discuss the recent XAIP literature enough
> can provide specific references or links to ones they think I should cover, as
> opposed to just the authors and years.